# Response of runoff towards land use changes in the Yellow River Basin in Ningxia, China

**Zhanping Wang[1,2,3], Juncang Tian [2,3,4] \*, Kepeng Feng[2,3,4]**

**1** School of Mathematics and Statistics, Ningxia University, Yinchuan, China, **2** Ningxia Research Center of Technology on Water-Saving Irrigation and Water Resources Regulation, Yinchuan, China, **3** Engineering Research Center for the Efficient Use of Water Resources in Modern Agricultural in Arid Regions, Yinchuan, China, **4** College of Civil and Hydraulic Engineering, Ningxia University, Yinchuan, China

\* wang_zp@nxu.edu.cn

**Data Availability Statement:** All relevant are within the paper and its Supporting information files.

**Funding:** The research has obtained the support from the Priority Research and Projects for Ningxia in China (2020BEG03021), the First-class Major

## Abstract

Since the Yellow River is a main source of water in Ningxia China, maintaining its healthy ecological environment is vital to Ningxia and the neighboring areas. Changes of land use caused by human activities such as population growth, urbanizing process, and industrial and mining construction would affect the balance and cycle of water in the Basin. Therefore, investigating hydrological responses of land use changes can provide insights into the characteristics and evolution of runoff the Yellow River Basin in the Ningxia section. This has imperative and practical significance to the rational use, allocation, and planning of water resources in a changing environment. In this paper, we analyzed the meteorological and hydrological elements of changing characteristics of the Yellow River Basin in the Ningxia section. Then we selected a distribution-based hydrology model of SWAT in combination with GIS to simulate annual and monthly runoff under different land use scenarios. Finally, we analyzed the influence caused by the changes of land use on runoff. We concluded that it is appropriate to lay a decision-making foundation to manage water resources of the Yellow River Basin in the Ningxia section.

## Introduction

With the population growth and rapid social and economic development in China, industrial, agricultural, domestic, and ecological water uses have increased dramatically, leading to an increasingly prominent contradiction between water supply and demand. Some irrational human activities have caused global warming problems. Other activities such as human reclamation of waters and destruction of grass and land have disrupted land use, caused serious soil erosion, and reduced vegetation coverage. Thus, the related regulation ability exhibited by the Yellow River Basin has decreased, resulting in frequent occurrences of natural disasters including droughts and floods. The issue of water resources has attracted attention of many researchers. The problem has become a bottleneck in the local economy development. Over the past few years, large numbers of scholars and researchers started investigating the impact caused by land use changes on water resources. Such research can lay a theoretical foundation for rational water use and development, buffet against natural disasters and soil erosion, and help

Foundation of Ningxia Institutions of High Education in China (NXYLXK2021A03), Natural Science Foundation of Ningxia Province (2020AAC03058, 2021AAC02007), the First-class Discipline Construction Project in Ningxia Universities: Mathematics, and the Scientific Research Project of Ningxia Colleges and Universities, China(NGY2020006).

**Competing interests:** The authors have declared that no competing interests exist.

decision-makers make relevant policies in relation to the River Basin. Land use refers to the natural attribute of human use and the actual existence of land. Research generally focuses on the temporal change trend of its quantity and structure and its spatial distribution and use types. Land use changes are simultaneously affected by natural conditions and socio-economic development. It is a relatively complex change process.

The hydrological model is a tool for evaluating future changes in hydrology and water resources. For instance, the model of SWAT (Soil and Water Assessment Tool) is extensively employed in the Baltic Sea [1, 2]. Land use changes significantly impact hydrology and water resources [3]. More than one third of land surface is altered by human activities [4]. Such activities will continue following an increasing demand for human resource [5]. From 1990 to 2015, the global forestland decreased by 3%, and most of the reduced forestland was converted into agricultural land [6]. Land use/land cover (LULC) changes, e.g., wetland reclamation, urbanization, desertification, and afforestation/ deforestation, led to an increase in droughts and floods, and land degradation associated with erosion of soil caused a decline in agricultural productivity, making the natural ecosystem keep deteriorating [7]. Land use and land cover significantly help maintain the ecosystem structures and productivity [8]. Numerous LULC categories offer considerable numbers of ecosystem services [9]. Meanwhile, changing LULC has a great impact on ecosystem services and the pattern of landscape [9, 10]. Large-scale changes in LULC are likely to make natural habitats to be fragmented or landscape pattern homogenized [11]. Therefore, the impact caused by land use changes on hydrology process and sediment production should be investigated. This can offer essential data to develop the management of water resources and strategies for land use planning.

In China, the Yellow River is notable because of the comparatively low water production in contrast to its significant sediment production. The latter takes up approximately 6% of the world's sediment production in all river mechanisms [12]. Nevertheless, the recent sediment and runoff have been highly reduced inside the Yellow River Basin though some abrupt periods occurred in the late 1980s and the early 1990s [13, 14]. Human interventions such as taking soil and water protection measures compounded with climate change will cause a significant reduction of runoff and sediments inside the middle reaches of the Yellow River [15]. The Loess Plateau, with the location inside the middle reaches of the Yellow River's, takes up nearly 380,000 sq.km. Water in that area takes up 44.3% of the Yellow River's flow while sediments take up 88.2% of river sediments [15]. Overall, the Loess Plateau region turns out to be the most serious soil erosion area worldwide.

A study by Jennings and Jarnagin [16] revealed that in Virginia, USA, the streamflow response to land use and land cover change increased noticeably with the impervious areas sharply rising from 3% to 33%. Kim et al. [17] and Beighley et al also obtained similar findings. Niehoff et al. [18] indicated that in a meso-scale watershed in southwestern Germany, the impact of land use status on storm runoff generation is largely dependent on the features of rainfall events and relevant spatial scales. That is, the effect is only associated with convective storm events with a high intensity of rainfall, and is opposite to long-term convective storm events with a low intensity of rainfall. These results manifest that the time and space scales, climate variations, and physical features of the study area can all have decisive impacts on hydrology. Therefore, it is important to conduct investigations on this issue with a more systematic and comprehensive manner. Wagener [19] pointed out that there are still controversies in the hydrological impacts of land use and changes in land cover and further research is necessary. To quantify the effect of changes in land use on hydrological elements on a watershed scale, a hydrological model is needed [20–23].

Some researchers have studied the influence of climate and changes of land use on hydrology inside various regions using the SWAT model [24–27]. The SWAT model is used to

simulate the influence caused by different land use types on water resources over time [28]. Zuo et al. [29] employed statistical methods, hydrological models, and land use maps to explore the effects of land use and climate change on water resources and sediments within the Huangfuchuan watershed of the Loess Plateau. Based on the SWAT model, the general and specific effects from climate and land use changes in the East Baltic Sea area on future river runoff were evaluated [30]. However, there is limited research applying the SWAT model to examine the changes in daily runoff. Generally, the study of hydrological changes only focuses on annual runoff but this cannot fully cover daily changes. Though Zhou et al. [26] simulated yearly, monthly, and daily runoff, time scales are rarely discussed in the hydrological change literature, and there are little discussion about what time indicators are suitable for daily runoff. Compared with long-term runoff, short-term runoff is less stable and has a more complicated relationship with land use. Thus, it is meaningful to investigate the influence of land use upon runoff at different time scales for hydrological processes. In the present study, by using SWAT, we simulate the changes within annual and monthly runoff in various land use conditions of the Yellow River Basin in the Ningxia section. We also analyze the differences between the time indicators.

## Materials and methods

In this paper, the influence of land use changes on runoff in the watershed of the Loess Plateau was first studied. Then, the land use transfer matrix was selected for an analysis of the changes of land use within the Yellow River Basin in the Ningxia section. Finally, the influence of changes of land use upon the runoff pertaining to the watershed was analyzed using the calibrated SWAT model. This research could lay a scientific foundation for planning of water resources, management, and soil and water conservation within the Loess Plateau.

### Overview of the research area

In Ningxia, the Yellow River starts from Xiaheyan in Zhongwei City and flows through Huinong District, Dawukou District, Pingluo County, Helan County, Yinchuan City, Yongning County, Lingwu City, Litong District, Qingtongxia City, Zhongning County, and Shizuishan City, then extends to Inner Mongolia. Three hydrological stations have been established along the Yellow River mainstream, which are Shizuishan, Qingtongxia, and Xiaheyan stations. In Ningxia, the Yellow River Basin has the location at 36°0′ ~ 39°23′ north latitude and 104°17′ ~ 107°39′ east longitude, covering an irrigation area of 51400 km$^2$. An alluvial plain refers to the landform type, with flat terrain, vertical and horizontal ditches, and an altitude of 1100–2500 m; it refers to a sufficient sunshine based temperate arid area, and the annual sunshine hours on average in the basin ranges between 2750 and 2950 h; the annual wind speed on average ranges between 1.7 and 2.5 m/s. Besides, it presents a large temperature difference, rich heat, as well as a relatively long frost-free time. It has less rainfall but more droughts with a typical continental climate. Annual rainfall is 180–220 mm and uneven distribution of precipitation is all year around; the seasons are distinct for wet and dry conditions; the precipitation from July to September takes up 60%~70% of the yearly precipitation. The land consists of cultivated land and grassland, taking up 48.76% and 33.74% of the Basin area, respectively. Forest land takes up 3.6%, water area 1.43%, rural and urban industrial and mining residential land 1.46%, and unused land 11.01%.

### Application of SWAT

The model of SWAT (Soil and Water Assessment Tool) has a sub-module which resembles connections of the water cycle and presents advanced designing ideas. This designing

approach facilitates the model to be expanded and applied. The model builds a crucial link between basin processes and land use activities, enabling it to evaluate the applicability of a wide range of determinations in a basin management. Moreover, different sub-models can be selected according to research aims. The models exhibit special characteristics for an operating one. The SWAT model employs a particular command code control approach for controlling water flow evolution between subbasins and in the river network.

SWAT divides the research basin into multiple subdivided-basins in accordance with the river network and water mechanism, maintaining the geographical location of the basin and the spatial relationship between the subbasins. Each subdivided-basin has a wide range of meteorologically and hydrologically related, land use, soil, agriculture managing measures, and pesticide use. Subsequently, the subbasin falls to many units of hydrologic response (HRUs) according to elements, e.g., land use type and soil category and their normal attributes. The HRU acts as the smallest unit for hydrology simulation for simulating the different hydrology cycle components as well as their quantitative transforming relationships. Next, the non-point source load, runoff generations, and sedimentation pertaining to the respective HRU within the subbasin receive the summary. Lastly, the relationship of the basin's water balance is determined by calculating river network confluence.

The model of SWAT conducts the simulation of the water cycle's land phase with the water balance expression below:

$$SW_t = SW_0 + \sum_{i=1}^{t} \left( R_{day} - Q_{surf} - E_a - w_{seep} - Q_{gw} \right) \tag{1}$$

in the formula, $SW_t$ denotes final soil water content (mm); $SW_0$ represents initial soil moisture (mm); $t$ expresses time Step (day); $R_{day}$, $Q_{surf}$, $E_a$, $w_{seep}$ and $Q_{gw}$ represent infiltration, lateral flow, evaporation, surface runoff and precipitation of soil aquifers, as well as groundwater content in the i-th day (mm), respectively.

Prior to the simulating process based on the SWAT model, relevant database files, land use and soil maps, related digital elevation should be prepared for facilitating the set-up of the SWAT model input information. Given artificial disturbance, e.g., river courses, drainage ditches, and irrigation canals within plain irrigation areas, the SWAT model receives the optimization regarding the extracting approach of ditch and river network, the subbasin, and HRU division module. A specific method is detailed as follows. In accordance with the space distributing characteristics pertaining to an artificial drainage ditch network and natural river channels within the study area, the "burn-in" algorithm is adopted for denting the DEM. Such an approach primarily aims at transforming the drainage ditch network and digital river system in a grid form. The size of the grid cell is the same as the original DEM. It receives the transformation to a unified coordinate mechanism based on projection and subsequent superimposition on the original DEM via superposition operation. The elevation value pertaining to the raster unit where the corrected river course is located remains unchanged, and the elevation value of the grid unit where it is perpendicular to the direction of the river course is increased by a small value; the increment is based on the elevation of the river grid lower than the elevation of the coast. Afterward, the modified DEM is reprocessed for generating a novel river basin network. In terms of man-made water distributing channels, the DEM receives the processing based on the algorithm of elevation increment superposition. For making the subbasins divided by the SWAT model in the irrigation area conforming to the irrigation areas covered by the trunk and branch canal system and to avoid the model generating too many subbasins in the process of spatial discretization (resulting in excessive spatial data conversion and calculation), a reasonable threshold for a minimum waterway catchment area is set and

divides the subbasin with the area covered by the backbone channels and the main drainage ditches as the dividing standard. Subsequently, each subbasin is divided in accordance with the matching of land use and soil type, and the farmland area is refined to form a hydrological response unit under the consideration of different crop planting structures.

The digital assessment model shows the indispensable property if applying SWAT model to hydrological process simulation, water system generation, as well as basin division. The most common DEM diagram is a grid type. The application of DEM data is capable to calculate the slope and slope length parameters belonging to the respective subbasin and define the river network of the basin. The characteristics of river slope, slope length, and width of the river basin network are all extracted from the DEM data.

DEM is a relatively smooth terrain surface model. However, there are some depressions on the surface of the DEM due to errors and some special topography. Consequently, low-precision water flow direction results are obtained, making the original DEM data fail to satisfy the needs of the research. Accordingly, prior to most of the simulation experimental processes, the original DEM data can receive the filling process based on the hydrological analysis model of ArcGis software to obtain the DEM data without depressions to meet the research needs.

The high-resolution DEM data (SRTM: Shuttle Radar Topography Mission 90-meter resolution data) employed in this study are available in the Chinese Academy of Sciences' (http://datamirror.csdb.cn/index.jsp). International Scientific Data Service Platform and modified to meet this study needs (Fig 1). (Republished from [Institute of Geographic Sciences and Natural Resources Research of Chinese Academy of Sciences] under a CC BY license, with permission

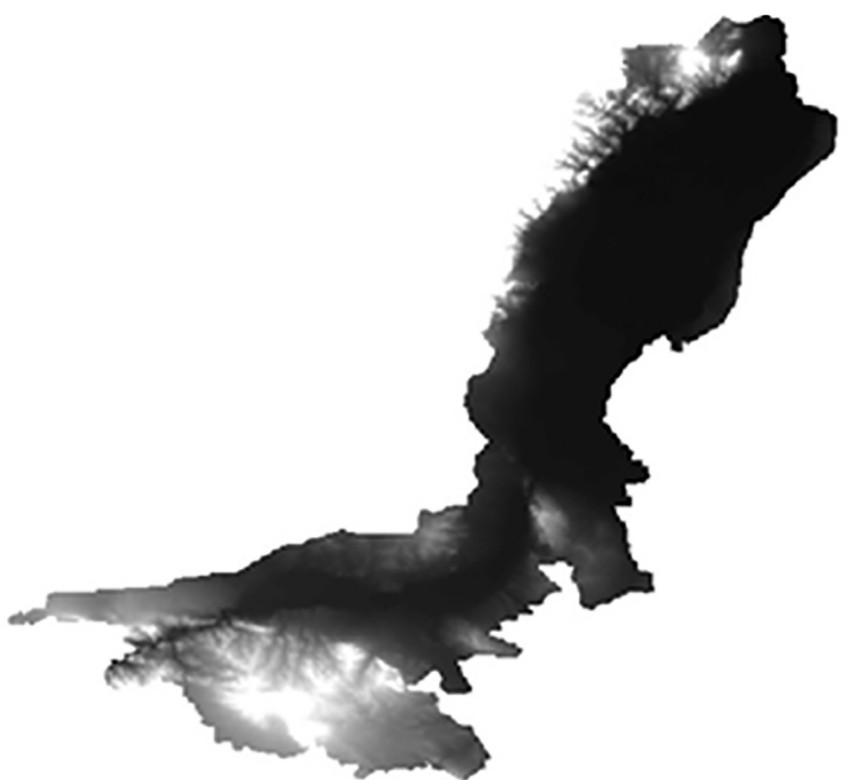

**Fig 1. DEM map of Ningxia section of the Yellow River Basin.**

from [Resource Environmental Science and Data Center of Chinese Academy of Sciences], original copyright [2004]).

Land use maps are useful when applying the SWAT model for simulating water resources within a river basin. It is not easy for humans to alter the topographies and soil within the watershed whereas the land use could be altered through the efforts of human. Thus, an application of the latest land use map is recommended for the simulating process. In terms of the projection coordinate system of the land use distribution map, it needs to be transformed by the Projections of ArcToolbox if it varies from the setting in the research. The types defined in the land use map shall be classified into two levels based on the city data base and the SWAT land cover/vegetation type data base, and a query form in the required format shall be developed. This makes it easy to determine the code (4 characters required) in the SWAT model for each land use type on the land use map.

The land use maps collected in this paper are in a SHP format. It was first cut according to the boundary of the watershed and then converted into a Grid format. Because the land use classification system of the SWAT model aligns with the classifying standards of the United States, which is different from those of China, the land use types were reclassified and converted into the land use map into the codes that can be recognized by the SWAT model. Using the Classification Standard of Land use Status (GB/T21010-2007) and the land use attribute data base coming from the SWAT model, we established a land use data base in line with the Yellow River irrigation region. Table 1 lists the reclassified land use types.

Table 1. Reclassification code conversion of land use types.

| Original classification and coding | | Reclassification and coding | | | Area (km$^2$) | Percentage of total area (%) |
|---|---|---|---|---|---|---|
| No. | Land use type | Coding | Type in SWAT | SWAT code | | |
| 11 | Irrigated paddy fields | 1 | Agricultural Land Generic | AGRL | 3750 | 45 |
| 12 | Dry land | | | | | |
| 21 | Wood land | 2 | Forest-Mixe | FRST | 217 | 2.6 |
| 22 | Shrubbery lands | | | | | |
| 23 | Sparsely forested woodland | | | | | |
| 24 | Other forestland | | | | | |
| 31 | Natural grassland | 3 | Pasture | PAST | 1117 | 13.4 |
| 32 | Improved grassland | | | | | |
| 33 | Man-made grassland | | | | | |
| 41 | River surface | 4 | Water | WATR | 708 | 8.5 |
| 42 | Lake surface | | | | | |
| 43 | Reservoir surface | | | | | |
| 46 | Beaches and flats | | | | | |
| 64 | Wet land | | | | | |
| 52 | Residential quarters in rural areas | 5 | Urban | URBAN | 393 | 4.7 |
| 51 | Areas of cities and town | | | | | |
| 53 | Specially-used land | | | | | |
| 61 | Sandy land | 6 | Bare | BARE | 2154 | 25.8 |
| 62 | Gobi | | | | | |
| 63 | Saline-alkali land | | | | | |
| 65 | Bare land | | | | | |
| 66 | Exposed rock and shingle land | | | | | |
| 67 | Others | | | | | |

After loading the land use look-up table file in the SWAT model, the reclassified land use types are illustrated in Fig 2. (Republished from [Institute of Geographic Sciences and Natural Resources Research of Chinese Academy of Sciences] under a CC BY license, with permission from [Resource Environmental Science and Data Center of Chinese Academy of Sciences], original copyright [2004]).

## The impact of land use changes on watershed runoff

There are links and differences between land use and land cover. Land use refers to the land resource used by human activities to obtain products or services they need. Humans change the land cover through the activities of land use, which is one of the main ways that human activities affect the earth system. Land use changes have a crucial impact on biodiversity, regional water quantity and quality, and environmental adaptability, leading to changes in the earth's biogeochemical cycle. Land use focuses on the socio-economic attributes of land. Land cover refers to a complex of various elements on the ground covered by natural structures and artificial buildings. Its shape and state can change over time and space. Land use is mainly determined by natural factors such as hydrology, climate, and geomorphology, and has specific time and space attributes. Changes in land cover are caused by human land use activities. Land cover focuses on the natural attributes of the land. Land use/land cover changes can be divided into two categories: transformation and mutation. Transformation indicates the change from one type of land cover to another, such as the conversion from construction land to non-construction land. Mutation refers to the change between land cover types, such as the change from agricultural land to industrial land.

With in-depth research of global changes, the impact of human activities on environmental changes is recognized by the majority of scientific researchers. As the social economy develops, human beings continue to develop and use land, causing changes in land cover. These two changes have become the main factors of global environmental changes. China's current research on land use is also imminent. With the rapid development of China's social economy, the pace of human using and developing land resources is accelerating. This leads to conditions such as water and soil erosion and pollution, and this ultimately results in serious degradation of land quality. Besides, the per capita land occupation has gradually decreased with the increase in population, resulting in a shortage of land resources. The use of land resources is closely related to the use of water resources. The distribution of water and land resources in China is extremely uneven, and especially in areas with severe water shortages, water resources are vital to land use.

## The impact of land use changes on hydrology and water resources

LULC changes in a watershed will have varying degrees of impact on each link of the watershed hydrological cycle. The deeper impact is reflected in the change in water quality and quantity. LULC changes affect the water cycle, quality, and quantity by changing the surface evaporation soil moisture status and the interception amount of the ground cover, which, in turn, affects the water balance of the basin. Land use changes have altered the structure of ground vegetation, causing changes in water cycle factors such as evaporation, soil water content, runoff, and infiltration. Consequently, the regional water conversion is significantly affected. LULC changes are one of the main driving factors of hydrological response. Overseas researchers emphasize the impact of land use on runoff and water quality. For example, Bormann et al. investigated the effects of different land use methods on soil and water quality [31]. Chinese researchers have adopted similar directions, mainly simulating the changes in runoff under the change of land use patterns. For instance, Pang Jingpeng et al. simulated and

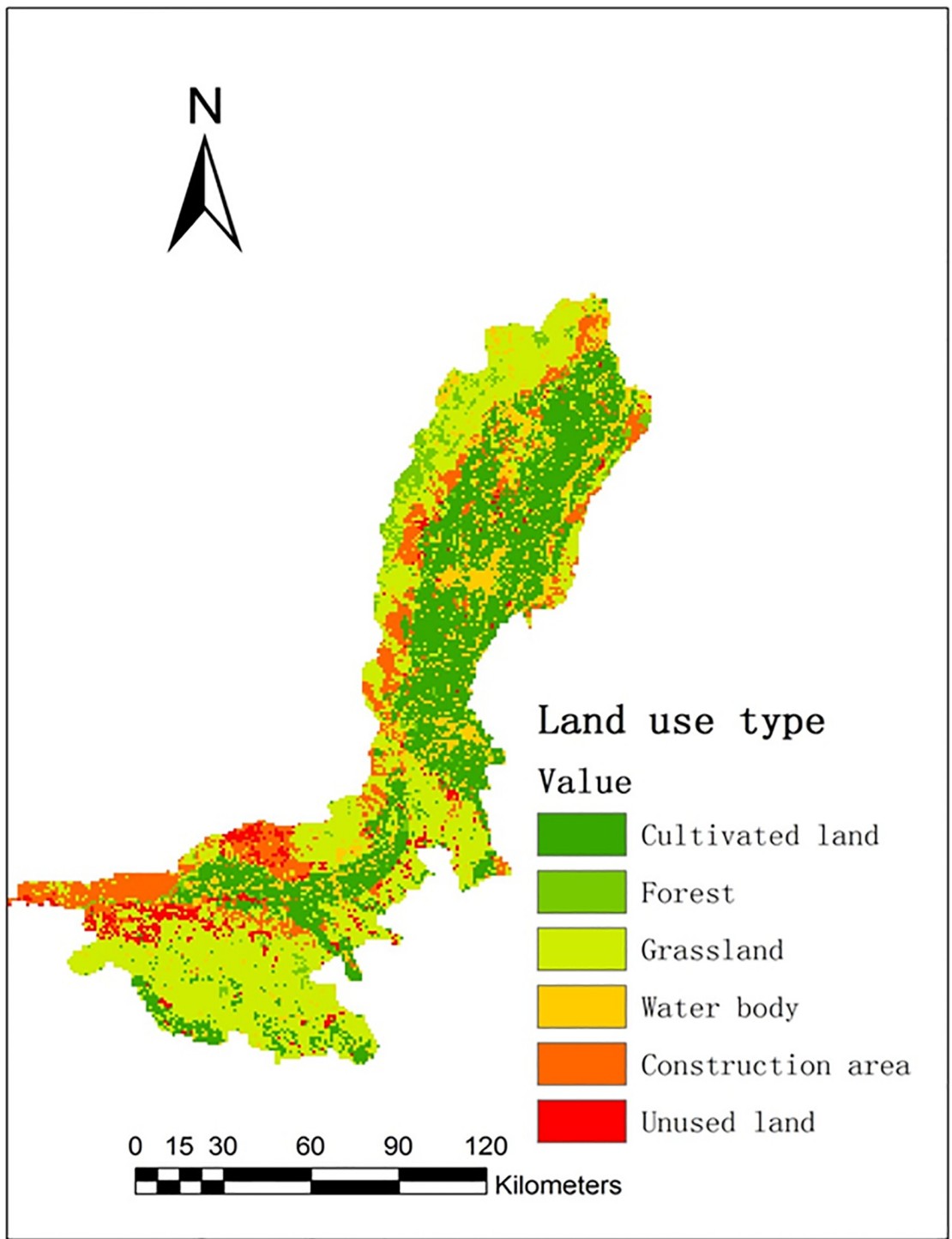

**Fig 2. Land use map of Ningxia Yellow River irrigation region in 2004.**

compared the runoff and sediment volume of the Miyun Reservoir basin based on land use in different periods. Land cover changes caused by human farming and resettlement (urban sewage) have caused worldwide water pollution.

## Scenario designs for land use changes

Land use involves the underlying surface of the watershed. The changes will have a significant impact on the amount of water resources and hydrological processes in the basin. According to the actual situation of the study area (considering various factors), different land cover scenarios in the studied area were established, and these are simulated using the SWAT model. The simulation results were employed to analyze the changes in runoff under different land cover scenarios to reveal the impacts of land use changes on runoff in the watershed. For the Ningxia section of the Yellow River Basin, land use types primarily include cultivated land, woodland, grassland, and bare land. Besides, the economic development of the Yellow River irrigation area is considered at the same time to establish the following three simulation scenarios:

Scenario 1: Given the actual situation in the Ningxia section of the Yellow River Basin, the policy of returning farmland to forest and grassland, and the implementation of sand control projects, the grassland area remains unchanged, and the existing arable land area has reduced by 10% in simulation. All the reduced arable land will be converted into forestland. Hence, the area of forestland has increased by 10%. The area of other types of land remains unchanged. The changes in runoff under this scenario are investigated.

Scenario 2: Considering the actual situation of the watershed and the small changes in the forest area in recent years, the forest area remains unchanged in this scenario, and the existing cultivated land area has reduced by 10% in simulation. The reduced arable land is all converted to grassland. That is, the grassland area has increased by 10%. The area of other types of land remains unchanged. The changes in runoff under this scenario are investigated.

Scenario 3: The "Thirteenth Five-Year Plan" significantly contributes to the development of animal husbandry. As the number of livestock continues to increase, the area of pasture will continue to decrease. Taking the influence of this factor into account, the area of other types of land remains unchanged in this scenario, and the existing grassland area has reduced by 10% in simulation. All the reduced grassland is converted into forestland.

That is, the area of the forestland has increased by 10%. The changes in runoff under this scenario are researched.

The percentage of the land use area in the above three scenarios is presented in Table 2.

## Results and discussion

According to the above three hypothetical land use scenarios, the relevant data of land use has changed using the SWAT model to simulate the runoff under different scenarios and the average monthly runoff. The simulation results are illustrated in Table 3 and Fig 3.

**Table 2. Land use area of various scenarios in the basin.**

| Land use type | Cultivated land (%) | Forest (%) | Grassland (%) | Water body (%) | Construction land (%) | Unused land (%) |
|---|---|---|---|---|---|---|
| 2004 | 45 | 2.6 | 13.4 | 8.5 | 4.7 | 25.8 |
| Scenario 1 | 35 | 12.6 | 13.4 | 8.5 | 4.7 | 25.8 |
| Scenario 2 | 35 | 2.6 | 23.4 | 8.5 | 4.7 | 25.8 |
| Scenario 3 | 45 | 12.6 | 3.4 | 8.5 | 4.7 | 25.8 |

**Table 3. Runoff changes under different land use scenarios.**

| Items | Types of land use in 2004 | Scenario 1 | Scenario 2 | Scenario 3 |
|---|---|---|---|---|
| Runoff simulation value (Ten thousand m$^3$) | 9683.2 | 9428.3 | 9311.1 | 9902.2 |
| Change rate (%) | 0.00 | -2.63 | -3.84 | 2.26 |

It can be observed from Table 3 that under different land use scenarios, the changes in runoff are:

1. For scenario 1, the area of land having received the cultivation has reduced by 10%. The area of other types of land remains unchanged when all the reduced cultivated land is converted into forestland. The simulated runoff is 9428.3 ten thousand m$^3$, which is 254.9 ten thousand m$^3$ less than that of the original land use type, with a decrease rate of 2.63%. This demonstrates that the runoff has decreased, the area of forestland has increased, and the area of cultivation land has decreased.

2. For scenario 2, the area of land used for cultivation has reduced by 10%. The area of other types of land remains unchanged when all the reduced cultivated land is converted to grassland. The simulated runoff is 9311.1 ten thousand m$^3$, which is 372.2 ten thousand m$^3$ less than that of the original land use type, with a decrease rate of 3.84%. This suggests that the runoff has decreased, the area of grassland has increased, and the arable land area has decreased.

3. For scenario 3, the grassland area has reduced by 10%. The area of other types of land remains unchanged when all the reduced grassland becomes forestland. The simulated runoff is 9902.2 ten thousand m$^3$, which is 219 ten thousand m$^3$ over that of the original land use type, with an increase rate of 2.26%. This indicates that the runoff has increased, the grassland area has increased, and the area of cultivated land has decreased.

Based on the mentioned situations, cultivated land experienced a reduction of 10%, which was all converted into forest land. The runoff was 9428.3 ten thousand m$^3$, showing a decrease of 254.9 ten thousand m$^3$ and a rate of 2.63%. Cultivated land experienced a reduction of 10%, which was all converted into forest land. The runoff was 9311.1 ten thousand m$^3$, showing a decrease of 372.2 ten thousand m$^3$ and a rate of 3.84%. Grassland experienced a reduction of 10%, which was all converted into forestland. The runoff was 9902.2 ten thousand m$^3$, showing a decrease of 219 ten thousand m$^3$ and a rate of 2.26%. The order of yield and discharge of distinct land use types in the Ningxia section of the Yellow River Basin is that cultivated land is bigger than forestland, which is bigger than grassland in terms of size.

As illustrated in Fig 3, specific to the three distinct land use scenarios, the significant change of runoff during June—September indicates that the runoff changes substantially during the flood season. Notably, the change reach the maximum from July to August and the minimum from December to February. The changes in runoff during flood seasons are relatively large, with the largest change in summer runoff and the smallest change in winter runoff, demonstrating that the land use changes within the watershed is more sensitive to runoff during flood seasons in summer. As shown from the Fig 3, the order of yield and discharge of distinct land use types in the Ningxia section of the Yellow River Basin is that cultivated land is bigger than forestland, which is bigger than grassland in terms of size.

## Conclusions

In this paper, the scenario analysis approach and the SWAT distributed hydrological model were employed to simulate the impact of land use changes on runoff in the Yellow River

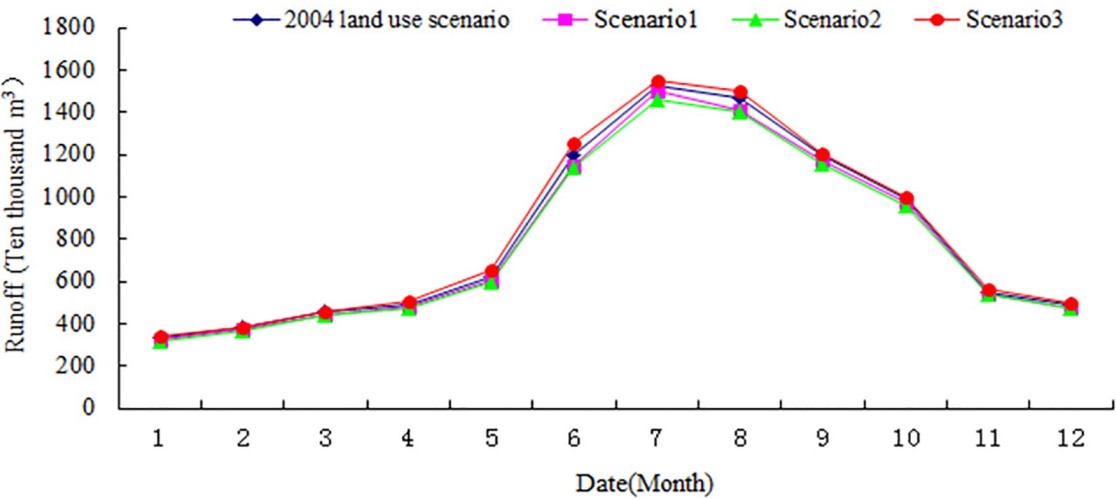

**Fig 3. Comparison of runoff under different land use scenarios.**

Irrigation region of Ningxia, China under the three different scenarios. Specifically, the impact level of land use changes on runoff is quantitatively distinguished by comparing and analyzing the simulated runoff values between different combination scenarios. A comparative analysis of simulation scenarios of analysis was conducted examining the influence caused by land use changes on runoff. The simulation analysis result reveals the order of the yield and discharge of different land use types. That is, cultivated land is bigger than forestland, which is bigger than grassland in size. Land use changes in the watershed are more sensitive to runoff during the flood season in summer.

## Supporting information

**S1 Data.**
(XLSX)

## Acknowledgments

We would like to appreciate the Yellow River Conservancy Committee for providing valuable data and Meteorological Department with data collection.

## Author Contributions

**Investigation:** Zhanping Wang.

**Methodology:** Zhanping Wang.

**Resources:** Juncang Tian.

**Software:** Juncang Tian.

**Supervision:** Kepeng Feng.

**Validation:** Kepeng Feng.

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
