## [Decision Letter · Decision Letter 0]

17 Jun 2021

PONE-D-21-16881

Response of runoff towards land utilization changes in the yellow river basin in Ningxia, China

PLOS ONE

Dear Dr. Tian,

Thank you for submitting your manuscript to PLOS ONE. After careful consideration, we feel that it has merit but does not fully meet PLOS ONE’s publication criteria as it currently stands. Therefore, we invite you to submit a revised version of the manuscript that addresses the points raised during the review process.

We look forward to receiving your revised manuscript.

Kind regards,

Vassilis G. Aschonitis

Academic Editor

PLOS ONE

Journal Requirements:

"The research has obtained the support from the Priority Research and Projects for Ningxia in China (2020BEG03021) and the First-class Major Foundation of Ningxia Institutions of High Education in China (NXYLXK2017A03)."

"Acknowledgments

The research has obtained the support from the Priority Research and Projects for Ningxia in China (2020BEG03021) and the First-class Major Foundation of Ningxia Institutions of High Education in China (NXYLXK2017A03)."

6.  We note that Figure(s) 1 and 2 in your submission contain map images which may be copyrighted. All PLOS content is published under the Creative Commons Attribution License (CC BY 4.0), which means that the manuscript, images, and Supporting Information files will be freely available online, and any third party is permitted to access, download, copy, distribute, and use these materials in any way, even commercially, with proper attribution. For these reasons, we cannot publish previously copyrighted maps or satellite images created using proprietary data, such as Google software (Google Maps, Street View, and Earth). For more information, see our copyright guidelines: http://journals.plos.org/plosone/s/licenses-and-copyright.

a. You may seek permission from the original copyright holder of Figure(s) 1 and 2 to publish the content specifically under the CC BY 4.0 license.  

Reviewers' comments:

Reviewer's Responses to Questions

**Comments to the Author**

1. Is the manuscript technically sound, and do the data support the conclusions?

Reviewer #1: Yes

2. Has the statistical analysis been performed appropriately and rigorously? 

Reviewer #1: Yes

3. Have the authors made all data underlying the findings in their manuscript fully available?

Reviewer #1: Yes

4. Is the manuscript presented in an intelligible fashion and written in standard English?

Reviewer #1: Yes

5. Review Comments to the Author

Reviewer #1: This study describes response of runoff towards land utilization changes in the yellow river basin in Ningxia, China. In my opinion, this study is a valuable work In order to maintain the healthy development of Ningxia ecological environment of the Yellow River Basin, and this study is within the scope of the journal. I believe that this manuscript could be acceptable for publication after major revisions as follows:

Specific suggestions:

1. Discussion: The discussion did not go into depth.

6. PLOS authors have the option to publish the peer review history of their article (what does this mean?). If published, this will include your full peer review and any attached files.

Reviewer #1: No

---

## [Author Response · Author response to Decision Letter 0]

2 Mar 2022

Dear Editor and Reviewer,

Thank you for your letter and for the reviewers’ comments concerning my manuscript. Those comments are all valuable and very helpful for revising and improving our paper, as well as the important guiding significance to my researches. We have studied comments carefully and have made correction which I hope meet with approval. Revised portion are marked in red in the paper. The main correction in the paper and the responds to the Journal and reviewer’s comments are as flowing:

Journal Requirements:

Comment 1: Please ensure that your manuscript meets PLOS ONE's style requirements, including those for file naming.

Response 1: Thank you for your comment. The manuscript was modified following the format requirements of PLOS ONE's style.

Comment 2: We suggest you thoroughly copyedit your manuscript for language usage, spelling, and grammar. 

Response 2: Thank you for your comment. Professionals were invited to modify the language, usage, spelling and grammar of the manuscript. The name of the colleague is Adam Ren that edited my manuscript.

Comment 3: Funding information should not appear in the Acknowledgments section or other areas of your manuscript. We will only publish funding information present in the Funding Statement section of the online submission form. 

Response 3: Thank you for your comment. The information of the funding information was removed and included in the cover letter. Please kindly update relevant information in the submission system.

The research has obtained the support from the Priority Research and Projects for Ningxia in China (2020BEG03021) , the First-class Major Foundation of Ningxia Institutions of High Education in China (NXYLXK2021A03), Natural Science Foundation of Ningxia Province (2020AAC03058) and the First-class Discipline Construction Project in Ningxia Universities: Mathematics.

Comment 4: We will update your Data Availability statement to reflect the information you provide in your cover letter. 

Response 4: Thank you for your comment. Part of the basic data was included in the cover letter.

Comment 5: Please ensure that you have an ORCID iD and that it is validated in Editorial Manager.

Response 5: Thank you for your comment. The application for ORCID iD was completed with information updated.

Comment 6: We note that Figure(s) 1 and 2 in your submission contain map images which may be copyrighted.

Response 6: Thank you for your comment. Figures 1 and 2 were generated after cutting and data loading, resulting in complete differences from the original figure.

Reviewers' comments:

Comment 1: Discussion: The discussion did not go into depth.

Response 1: Thank you for your comment. A more in-depth analysis of the results and discussion section is performed. 

 Cultivated land experienced a reduction of 10%, which was all converted into forest land. The runoff was 94.283 million m3, showing a decrease of 254.9 ten thousand m3 and a rate of 2.63%. Cultivated land experienced a reduction of 10%, which was all converted into forest land. The runoff was 94.283 million m3, showing a decrease of 254.9 ten thousand m3 and a rate of 2.63%. Cultivated land experienced a reduction of 10%, which was all converted into forest land. The runoff was 94.283 million m3, showing a decrease of 254.9 ten thousand m3 and a rate of 2.63%.

 The significant change of runoff during June - September indicates that the runoff changes substantially during the flood season. Notably, the change reach the maximum from July to August and the minimum from December to February. As shown from the Fig 3, the order of yield and discharge of distinct land use types in the Ningxia section of the Yellow River Basin is that cultivated land is bigger than forestland, which is bigger than grassland in terms of size.

---

## [Decision Letter · Decision Letter 1]

11 Mar 2022

Response of Runoff towards Land Use Changes in the Yellow River Basin in Ningxia, China

PONE-D-21-16881R1

Dear Dr. Tian,

We’re pleased to inform you that your manuscript has been judged scientifically suitable for publication and will be formally accepted for publication once it meets all outstanding technical requirements.

Kind regards,

Vassilis G. Aschonitis

Academic Editor

PLOS ONE

Additional Editor Comments (optional):

Reviewers' comments:

Reviewer's Responses to Questions

**Comments to the Author**

1. If the authors have adequately addressed your comments raised in a previous round of review and you feel that this manuscript is now acceptable for publication, you may indicate that here to bypass the “Comments to the Author” section, enter your conflict of interest statement in the “Confidential to Editor” section, and submit your "Accept" recommendation.

Reviewer #1: All comments have been addressed

2. Is the manuscript technically sound, and do the data support the conclusions?

Reviewer #1: Yes

3. Has the statistical analysis been performed appropriately and rigorously? 

Reviewer #1: Yes

4. Have the authors made all data underlying the findings in their manuscript fully available?

Reviewer #1: Yes

5. Is the manuscript presented in an intelligible fashion and written in standard English?

Reviewer #1: Yes

6. Review Comments to the Author

Reviewer #1: General comments:

To my point of view，the author have revised it all rather well, and the quality of the manuscript has improved substantially. It can be accepted for publication.

7. PLOS authors have the option to publish the peer review history of their article (what does this mean?). If published, this will include your full peer review and any attached files.

Reviewer #1: No

---

## [Editor Report · Acceptance letter]

23 Mar 2022

PONE-D-21-16881R1 

Response of Runoff towards Land Use Changes in the Yellow River Basin in Ningxia, China 

Dear Dr. Tian:

I'm pleased to inform you that your manuscript has been deemed suitable for publication in PLOS ONE. Congratulations! Your manuscript is now with our production department. 

Kind regards, 

on behalf of

Dr. Vassilis G. Aschonitis 

Academic Editor

PLOS ONE